# Temporal regulation of BMP2 growth factor signaling in response to mechanical loading is linked to cytoskeletal and focal adhesion remodeling

Sophie Görlitz[1,2,5], Erik Brauer[1,2,5], Rebecca Günther[1,2], Georg N. Duda[1,2,3], Petra Knaus [2,4] & Ansgar Petersen [1,2,3] ✉

Biophysical cues have the ability to enhance cellular signaling response to Bone Morphogenetic Proteins, an essential growth factor during bone development and regeneration. Yet, therapeutic application of Bone Morphogenetic Protein 2 (BMP2) is restricted due to uncontrolled side effects. An understanding of the temporal characteristics of mechanically regulated signaling events and underlying mechanism is lacking. Using a 3D bioreactor system in combination with a soft macroporous biomaterial substrate, we mimic the in vivo environment that BMP2 is acting in. We show that the intensity and duration of BMP2 signaling increases with increasing loading frequency in synchrony with the number and size of focal adhesions. Long-term mechanical stimulation increases the expression of BMP receptor type 1B, specific integrin subtypes and integrin clustering. Together, this triggered a short-lived mechanical echo that enhanced BMP2 signaling even when BMP2 is administered directly after mechanical stimulation, but not when it is applied after a resting period of ≥30 min. Interfering with cytoskeletal remodeling hinders focal adhesion remodeling verifying its critical role in shifting cells into a state of high BMP2 responsiveness. The design of biomaterials that exploit this potential locally at the site of injury will help to overcome current limitations of clinical growth factor treatment.

Bone tissue is highly mechanoresponsive and is constantly remodeled to meet the mechanical demands of daily life[1]. Bone strengthening in response to mechanical loading depends on the frequency, magnitude, and rate of loading[2–4] and is stimulated by cyclic rather than static loads[5]. Stimulation at a low magnitude and high frequency is particularly effective in promoting bone formation[6,7]. In addition to homeostatic bone remodeling processes, fracture healing is also sensitive to mechanical forces[8]. The mechanical boundary conditions at a fracture site, which are determined by the fixation system and in vivo loads, strongly influence the course and outcome of healing[9,10]. Based on in vivo observations and mechanobiological in silico analyses[11–13], researchers and clinicians aim for mechano-biologically optimized fracture fixations systems. In silico models of fracture healing are valuable tools to study the effect of different mechanical boundary condi-

tions. However, even though in silico models aim to account for the biological complexity, a comprehensive understanding of how cell behavior or even biochemical signaling pathways are influenced by mechanical forces is difficult to derive and in vitro experiments are required to study these complex processes.

There is strong evidence that biomechanical cues impact cells in their biochemical signaling and ultimately modulate cell behavior[14,15]. Bone Morphogenetic Protein 2 (BMP2) is known to play a central role in bone development and regeneration. Besides its indispensable role in the initiation of fracture healing[16] it gained considerable importance in the clinical treatment of tibial non-unions and spinal fusions after FDA approval of recombinant human BMP2 (rhBMP2)[17,18]. Interestingly, in vivo observations have indicated that mechanical forces and BMP2 jointly stimulate

[1]Julius Wolff Institute, Berlin Institute of Health at Charité, Berlin, Germany. [2]BIH Center for Regenerative Therapies (BCRT), Berlin Institute of Health at Charité, Berlin, Germany. [3]Center for Musculoskeletal Surgery, Charité - Universitätsmedizin Berlin, Berlin, Germany. [4]Freie Universität Berlin, Institute for Chemistry and Biochemistry, Berlin, Germany. [5]These authors contributed equally: Sophie Görlitz, Erik Brauer. ✉e-mail: ansgar.petersen@bih-charite.de

bone healing[19,20]. By varying the fixation stiffness, the bone-inducing effect of rhBMP2 can be even further enhanced when the mechanical load is high during the early bone healing phase but lower in later stages[19]. Tuning the mechanical environment by active application of axial compression during rhBMP2-induced defect healing resulted in a significant increase in mineralized tissue volume and mineral content 2 weeks after surgery compared to the control without axial compression[20]. These findings indicate that finely tuned mechanical conditions have the ability to enhance BMP2-induced bone healing. However, despite the relevance of BMP2 for both endogenous regeneration of sub-critical and therapeutic treatment of critical-sized bone defects, a comprehensive understanding of the interplay between mechanical forces and BMP2 signaling is lacking.

At the molecular level, in vitro studies have demonstrated a direct regulation of BMP signaling by mechanical forces in different cell types and experimental conditions[21–27]. Mechanical loading has been described to enhance the BMP-induced phosphorylation of receptor-associated Smad transcription factors (R-Smads), an immediate early event in BMP signaling, and downstream Smad target gene expression. There is strong evidence that this mechano-regulation is triggered by a crosstalk between mechanotransduction pathways and BMP signaling, but the molecular mechanism remains to be elucidated. Although different types of forces such as oscillatory fluid shear stress[24,26,27], cyclic tension[21], and compression[23,25] have been found to promote BMP signaling, systematic investigations on how different loading parameters and timing of mechanical stimulation influence the duration and strength of BMP signaling are lacking. Such studies are important, however, to define parameters that optimally support BMP signaling and, furthermore, to gain insights into the dynamics of this regulation.

The present study provides insights into the cellular events leading to mechanically regulated BMP signaling. We employ an in-house developed mechano-bioreactor to expose osteoblasts seeded in a 3D biomaterial environment to BMP2 stimulation and cyclic compression at different frequencies and durations. In addition to a frequency-dependent increase in BMP signaling, we show transcriptional co-regulation of specific integrin types and BMP receptor type 1B in response to long-term mechanical loading and BMP stimulation. This also resulted in a mechanical echo that enhanced BMP signaling, even when the cyclic compression was terminated. Importantly, we found that load-induced focal adhesion and F-actin remodeling are required for the amplification of BMP signaling by cyclic compression and the generation of the mechanical echo effect.

## Results

### Parameters of mechanical loading control BMP2 signaling response

To better understand the conditions resulting in enhanced BMP2 signaling in response to mechanical stimulation, the impact of the loading frequency on early BMP signaling events was investigated in a time-dependent manner. Cyclic biomaterial compression at a frequency of 1 Hz was chosen to mimic the dynamics of human locomotion[28]. This frequency was previously shown to enhance Smad1/5/8 phosphorylation[23]. Furthermore, 10 Hz, corresponding to muscle contraction[29], and an extremely low, rather non-physiological frequency of 0.03 Hz were also applied. Cell-seeded collagen scaffolds were transferred into the custom-made bioreactor (Supplementary Data 1A, B) and stimulated with BMP2 and cyclic mechanical compression (10% strain, $f$ = 0.03, 1 and 10 Hz). Figure 1c illustrates the scaffold wall and cell deformation under macroscopic compression indicating that cells are exposed to heterogeneous deformation patterns in the 3D environment mimicking the physiological situation. Cyclic compression was applied for 30, 90, or 120 min, and Smad1/5/8 phosphorylation was examined (Fig. 2a–d). Furthermore, the expression of the early BMP target

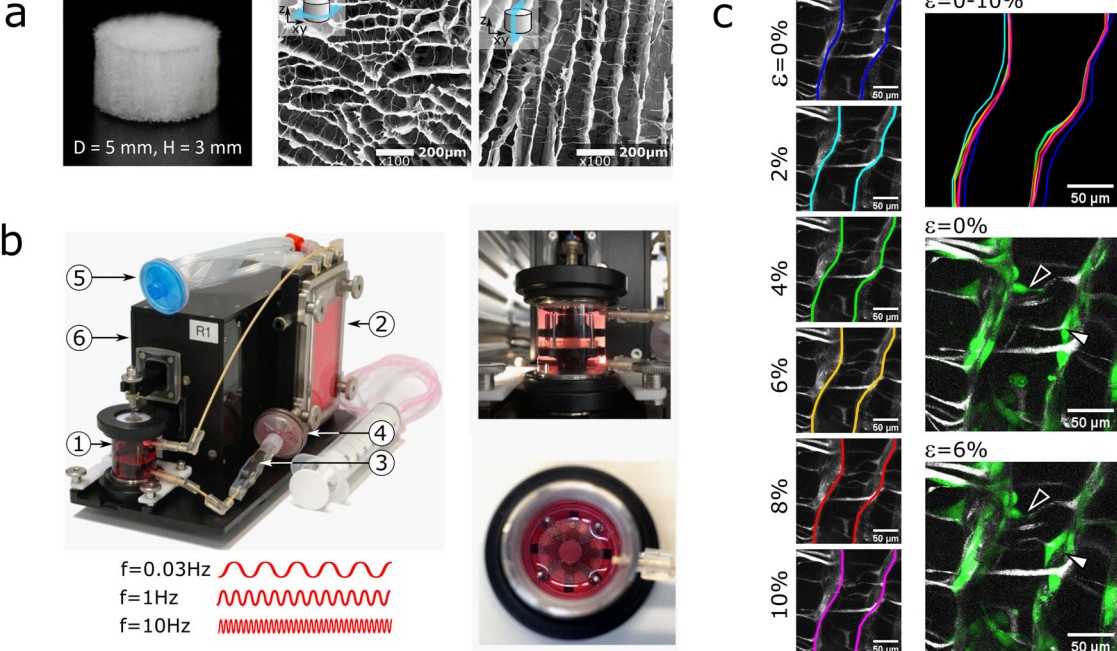

**Fig. 1 | Application of cyclic monoaxial compression to cells in 3D culture. a** SEM image of the macroporous collagen scaffold utilized in this study (axial and transversal cut). **b** Bioreactor consisting of the reactor chamber (1), medium reservoir (2), micropump (3), medium filter (4), sterile filter for gas pressure equalization (5), and the mechanical unit (6) first described by Petersen et al.[55]. **b** Close-up of the bioreactor chamber with collagen scaffold positioned between upper and lower plunger. Images on the right show the bioreactor chamber and the centrally placed biomaterial scaffold (side view of the closed chamber and top view of the open chamber after biomaterial placement). **c** Local biomaterial deformation under macroscopic compression was visualized via second harmonic imaging using an adapted version of the bioreactor (see Supplementary Data 1A, B). Left: Two scaffold walls are contoured exemplarily illustrating wall deformation at increasing compression up to 10% of sample height. Right: Overlay of the scaffold wall contours for 0–10% compression (top) and consequences of wall deformation on cell morphology (bottom) at low (0%) and high wall deformation (6%). Cell straining is heterogeneous and includes stretching/compression (white contour arrowhead) and bending with moving/deforming scaffold walls (white filled arrowhead). Scaffold walls via SHG in white, cells stained green using CellTracker™ (Thermo Fischer).

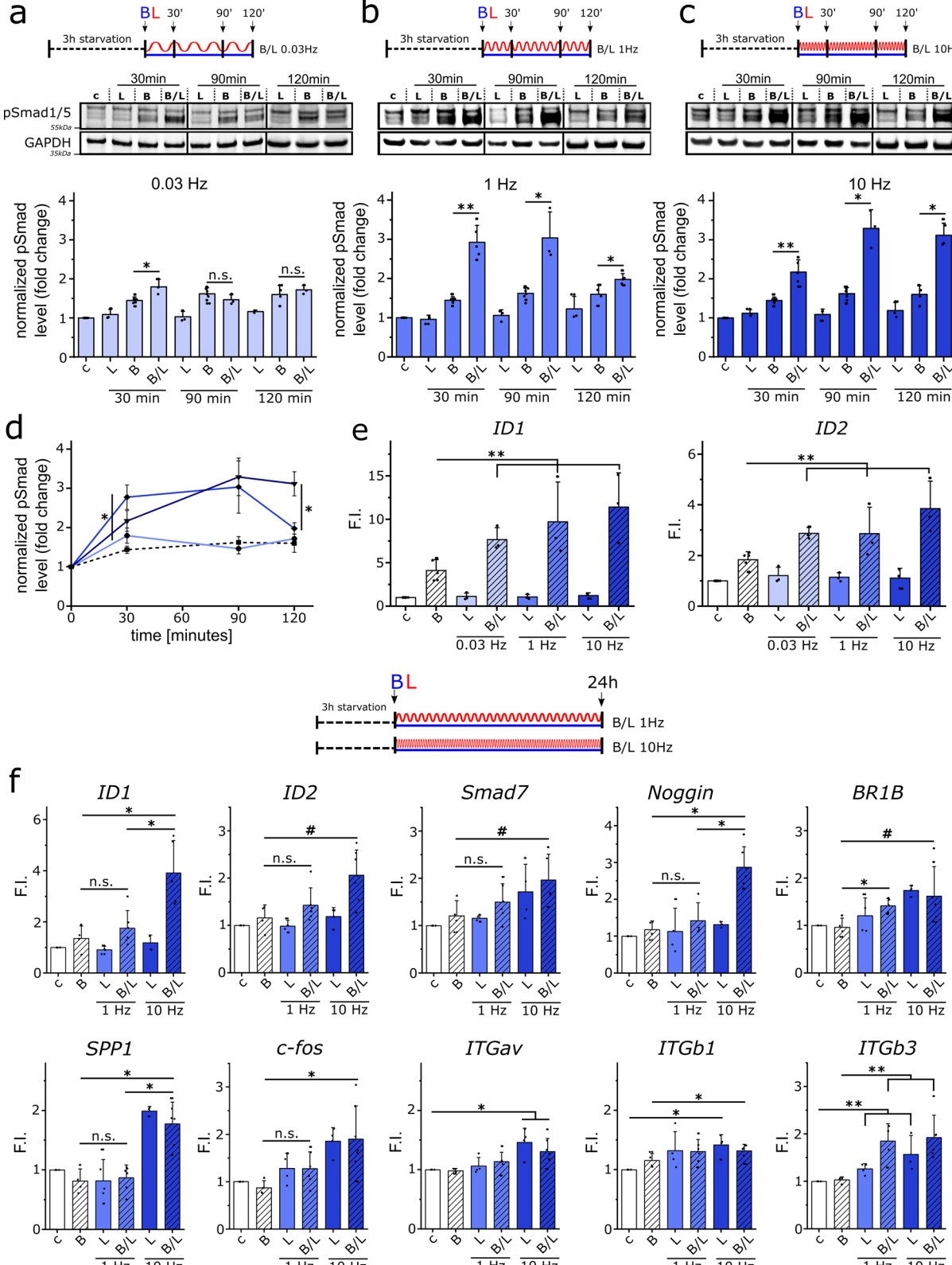

**Fig. 2 | Loading frequency influences strength and duration of Smad1/5/8 phosphorylation and gene expression.** Human FOBs seeded on collagen scaffolds were subjected to BMP2 stimulation, mechanical loading (10% compression), or a combination of both for 30, 90, or 120 min. Loading frequencies of 0.03 Hz (**a**), 1 Hz (**b**), and 10 Hz (**c**) were applied to analyze the impact on SMAD1/5/8 phosphorylation, determined using western blot analysis (**d**), and on (**e**) ID1 and ID2 expression (90 min), determined via quantitative qPCR

(relative to HPRT expression). F.I. = fold change relative to control samples. **f** Human FOBs seeded in collagen scaffolds were stimulated with BMP2 and/or mechanical loading (1 Hz or 10 Hz) for 24 h. Gene expression was analyzed by qPCR. Bar and line charts display means ± SD ($n ≥ 3$ from ≥3 independent experiments, #$p < 0.1$, *$p < 0.05$, **$p < 0.01$). Statistical significance was calculated using the two-sided Mann–Whitney U test and Bonferroni correction for multiple tests.

genes Inhibitor of DNA Binding 1 and 2 (ID1 and ID2) was analyzed by qPCR after 90 min (Fig. 2e).

Already after 30 min, simultaneous BMP2 stimulation and cyclic compression (B/L) at a frequency of 1 Hz and 10 Hz induced a similar significant increase in Smad phosphorylation in comparison to the sample treated only with BMP2. However, the increase in Smad phosphorylation induced by a loading frequency of 0.03 Hz, though significant, was much smaller compared to 1 Hz and 10 Hz. After 90 min of B/L stimulation, the maximum increase in Smad phosphorylation was reached at 1 Hz and 10 Hz loading frequency, whereas Smad phosphorylation dropped to the level of the BMP-only treated sample at 0.03 Hz. While Smad phosphorylation induced by B/L at a frequency of 1 Hz decreased to the level of the BMP-only treated control after 120 min, the maximum phosphorylation level was maintained with high-frequency loading at 10 Hz. The frequency-dependent phosphorylation of Smads was also reflected in the expression levels of ID1 and ID2 after 90 min. Especially ID1 transcription increased with increasing frequency. Interestingly, even B/L stimulation at a frequency of 0.03 Hz, which only mildly and transiently enhanced Smad phosphorylation, increased the ID1 expression two-fold. Taken together, these studies of the initial short-term co-stimulation effects demonstrated a strong and positive correlation between the frequency of cyclic compression and the duration of enhanced BMP signaling, which is indicated by a prolonged increase of Smad phosphorylation.

To further examine whether the initial frequency-dependent effects on Smad phosphorylation and early gene expression persist or equilibrates at a later time point, hFOBs were continuously stimulated with BMP2 and cyclic compression for 24 h. Loading at 0.03 Hz frequency was excluded from further studies due to its overall weak and non-persistent effect on early Smad phosphorylation, as no changes in gene expression were expected after 24 h for such a low and non-physiological loading frequency. The influence of cyclic compression at 1 Hz and 10 Hz with simultaneous BMP2 stimulation on the expression of direct BMP targets, osteogenic markers, and genes related to the perception of mechanical forces was investigated. The expression analysis indeed revealed a frequency-dependent increase in transcription even after 24 h (Fig. 2f). Direct BMP target genes ID1 and ID2 as well as the negative regulators of BMP signaling Noggin and Smad 7 were significantly increased by B/L stimulation at a frequency of 10 Hz compared to BMP treatment alone. In contrast, B/L stimulation at a frequency of 1 Hz had only minor effects. Interestingly, the expressions of BMP receptor type 1B, but not type 1 A or type 2, were significantly increased by simultaneous BMP2 stimulation and cyclic compression in a frequency-dependent manner, whereas BMP2 treatment alone had no effect (Supplementary Data 2). The expressions of the osteogenic marker genes RUNX2 and collagen type 1 α2 chain (COL1A2) were not affected by either treatment, but the expression of osteopontin (SPP1) was significantly increased by cyclic compression at a frequency of 10 Hz compared to the unstimulated control. The expression of c-fos, a transcription factor known to be a target of mechanotransduction[30], was increased with increasing frequency, reaching statistical significance at a frequency of 10 Hz.

Moreover, the expression of selected integrin subtypes was analyzed because integrins not only sense and transmit mechanical forces[31] but are also known to interact with BMP receptors and thereby influence BMP signaling[32–34]. Integrin αv, β1, and β3 expressions were found to be increased by mechanical loading, while α1, α5, and β5 were not affected (Supplementary Data 2). Cyclic compression induced integrin β3 expression in particular, which was further promoted by simultaneous BMP2 treatment, even though BMP2 treatment alone had no effect.

In summary, the frequency-dependent effects on early Smad phosphorylation persisted and transduced to the level of BMP target gene expression. The results revealed that mechanical loading at a frequency of 10 Hz significantly increases the duration of the crosstalk between mechanotransduction and BMP signaling in comparison to 1 Hz.

## Focal adhesion number and size are increased by BMP2 treatment and by mechanical loading in a frequency-dependent manner

Integrins provide a key mechanical link between the cell and the surrounding matrix and allow cells to sense and transmit substrate deformations[31]. Moreover, they have previously been described to modulate BMP signaling through their interaction with BMP receptors[32–34]. Focal adhesions (FA), in which integrins are clustered, represent hotspots of mechanosensation and transduction and could be important for the integration of mechanical signals into the BMP pathway. At the same time, FAs are acting as mechanical anchors between cells and the substrate and are thus transmitting any mechanical deformations of the surrounding ECM to the cytoskeleton. Therefore, we investigated whether the frequency-dependent increase in integrin expression under cyclic loading is converted into increased assembly of FAs. Human FOBs were stained for the focal adhesion marker phospho-Paxillin (pPax) after a 24-h treatment with BMP2 and/or cyclic compression (1 Hz or 10 Hz). Phospho-Paxillin represents a more selective marker of focal adhesion maturation compared to other markers such as integrins and vinculin. Yet, a strong co-localization of either of these markers with phospho-paxillin was observed, indicating that phospho-paxillin-positive FAs are generally enriched in integrins and vinculin (Supplementary Data 4). Quantifications of the number and size distribution of FAs in the 3D environment revealed a strong influence of the treatments on cellular adhesion to the walls of the collagen scaffold (Fig. 3a–c). While untreated cells exhibited only a few and small FA complexes, treatment with BMP2, cyclic compression, and a combination of both increased the number of FAs per cell significantly. The total number of FAs in cells treated with either BMP2 or 1 Hz cyclic compression was comparable and was not increased by a combination of both stimuli. However, mechanical stimulation at 10 Hz further increased the number of FAs in the cells with BMP2 treatment (Fig. 3b). Furthermore, the percentage of cells with FAs larger than 1 μm$^2$ increased about 1.5-fold under 1 Hz B/L treatment and about 2-fold under 10 Hz B/L treatment (Fig. 3c). Interestingly, stimulation with only BMP2 or only 1 Hz cyclic compression induced equal FA amounts and size distributions, even though BMP2 alone did not increase integrin expressions in contrast to 1 Hz cyclic compression.

The observed increase in integrin expression and FA assembly (Fig. 3d) together with the increased expression of BMP receptor type IB in response to cyclic compression without the application of BMP2 was particularly interesting, as both an increased amount of BMP receptors and the previously described BMP receptor–integrin interaction[27,35,36] could promote BMP signaling. It could be concluded that a 24 h stimulation with cyclic compression induces cellular adaptation processes that enhance Smad signaling once cells are exposed to BMP2.

## Mechanical enhancement of BMP signaling requires very timely loading events

Two questions arose from the observations described above:

1. Does long-term cyclic compression increase the overall sensitivity of cells to BMP2?

   As a result, the positive short-term effect of cyclic loading on BMP2 signaling would be further enhanced over time. To answer this question, cells were continuously mechanically pre-stimulated for 24 h before BMP2 was added. This pre-stimulation did not increase the levels of endogenously acting BMP (Supplementary Data 5) and we already excluded that inhibitory signaling cascades might influence the outcome as regulations either of BMP antagonist Noggin or of inhibitory Smad 7 was observed for 10 Hz frequency only (Fig. 2). Mechanical stimulation was continued for a further 90 min before sampling (Fig. 3a).

2. Does long-term cyclic compression induce sustained cellular sensitivity to BMP2 beyond mechanical loading?

   As a result, BMP signaling would be enhanced even if mechanical stimulation was discontinued. Cells were therefore mechanically stimulated for 24 h, then the stimulation was stopped and BMP2 was

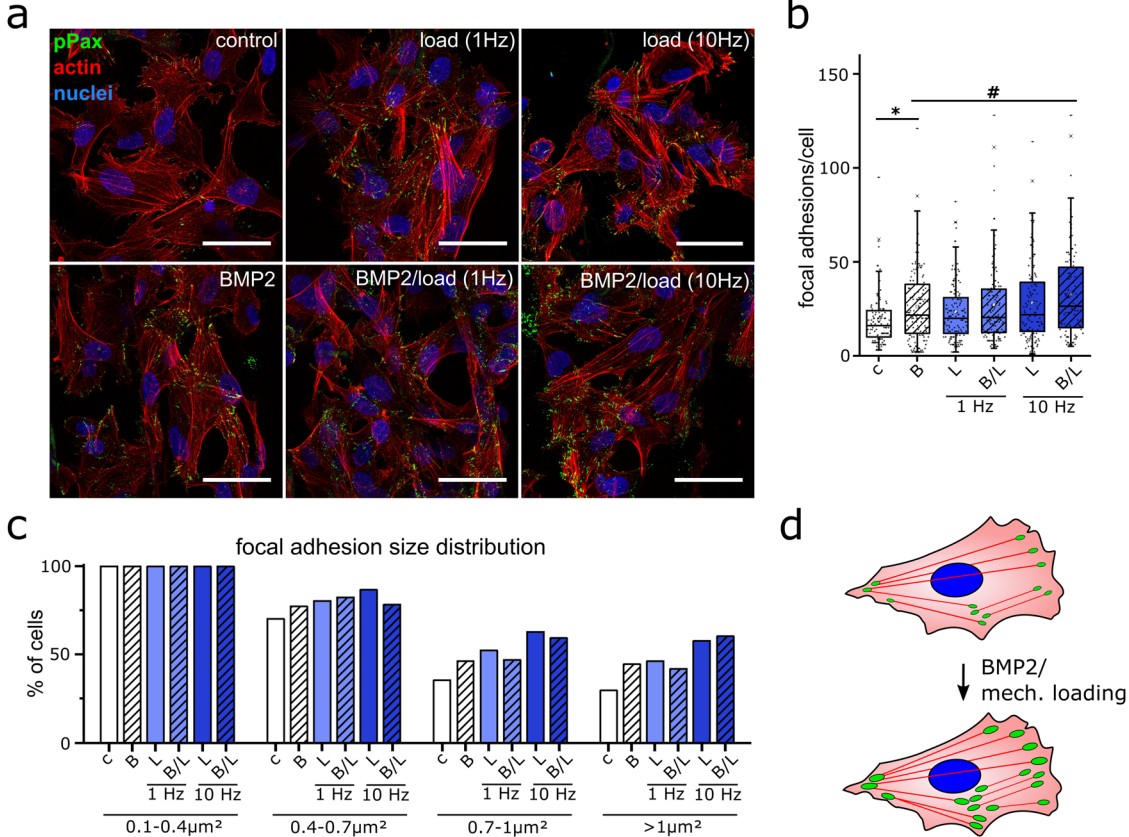

**Fig. 3 | Amount and size of focal adhesion are increased by BMP2 treatment and by mechanical loading in a frequency-dependent manner.** Human FOBs were seeded in collagen scaffolds, transferred into the bioreactor, and stimulated with BMP2 (5 nM) and/or mechanical loading (1 Hz or 10 Hz) for 24 h. Cells were fixated and stained for phospho-Paxillin (pPax, green), F-actin using phalloidin (red), and nuclei using DAPI (blue). **a** Representative confocal images of stained hFOBs. Scale bars represent 50 μm. **b** The amount of phospho-Paxillin positive FA per cell and **c** the percentage of cells with FAs of different size classes was assessed in ImageJ (in total >110 cells from at least 3 independent experiments). **d** Schematic illustration of the increase in FA size and amount due to BMP2 and mechanical stimulation. Boxplots in (**b**) show 25% lower and 75% upper box limits, median center line, and outliers as whiskers (>1.5× interquartile range). Statistical significance was calculated using a two-sided Mann–Whitney U test and Bonferroni correction for multiple tests (*$p < 0.05$, #$p < 0.1$).

added. No further mechanical stimulation was applied and the gene expression profiles of the cells were analyzed after 90 min of further culture in the bioreactor (Fig. 3a).

Gene expression analysis confirmed the previously observed significant increase in BMP receptor type 1B, integrin αv, and β3 expression by 24 h cyclic compression (Fig. 4b). However, to answer the first question, the 24 h mechanical pre-stimulation did not further promote Smad phosphorylation after BMP2 treatment (Fig. 4a).

Strikingly, Smad phosphorylation was increased to the same extent in mechanically pre-stimulated cells that did not receive further cyclic compression upon the addition of BMP as in samples treated with B/L for 90 min without mechanical pre-stimulation.

Even though Smad phosphorylation levels were equal in all three B/L samples regardless of pre-stimulation, the expression of BMP target genes ID1 and ID2 was reduced by mechanical pre-stimulation in comparison to only 90 min of simultaneous BMP and mechanical stimulation (Fig. 4b). This implies that intracellular negative regulators of the BMP pathway have been activated by mechanical stimulation downstream of Smad phosphorylation, which might interfere with the translocation of Smads to the nucleus or suppress the transcriptional activity of Smads.

When cells were rested for 30 or 120 min between mechanical pre-stimulation and BMP2 stimulation, p-Smad levels decreased slightly in comparison to the BMP-only control. To answer the second question, long-term mechanical pre-stimulation does not lead to a sustained activation of BMP2 signaling, but rather to a short, subsiding mechanical echo that

indicates rapid cellular adaptation processes after mechanical loading has ceased.

### The strength of the mechanical echo decreases with decreasing duration of mechanical stimulation

Based on the literature, it can be assumed that the cells have adapted to the changed mechanical environment within 24 h of cyclic compression. Non-transcriptional adaptations such as receptor reorganization and cytoskeletal adaptations, but also transcriptional responses including negative feedback mechanisms are initiated during this period until a new mechanical equilibrium is established[37]. We wanted to further understand how BMP signaling responds when the time of pre-stimulation is shortened so that cells do not have time to fully adapt to the altered mechanics at all levels, and thereby find out which cellular adaptation processes are responsible for the mechanical echo and the subsequent rapid decay of the crosstalk between mechanotransduction and BMP signaling.

The mechanical pre-stimulation was reduced to 90 and 30 min prior to 90 min BMP2 stimulation and phosphorylation levels of Smad1/5 were reanalyzed (Fig. 5a). Interestingly, the 90 min mechanical pre-stimulation was able to increase Smad1/5 phosphorylation in comparison to the BMP-only control. However, it was not as efficient, achieving only 50% of the increase induced by simultaneous B/L stimulation for 90 min, taking the BMP-only control as the baseline for a 0% increase in Smad phosphorylation (Fig. 5b). The 30 min mechanical pre-stimulation had no effect on BMP-induced Smad phosphorylation and was not sufficient to induce a mechanical echo affecting BMP signaling.

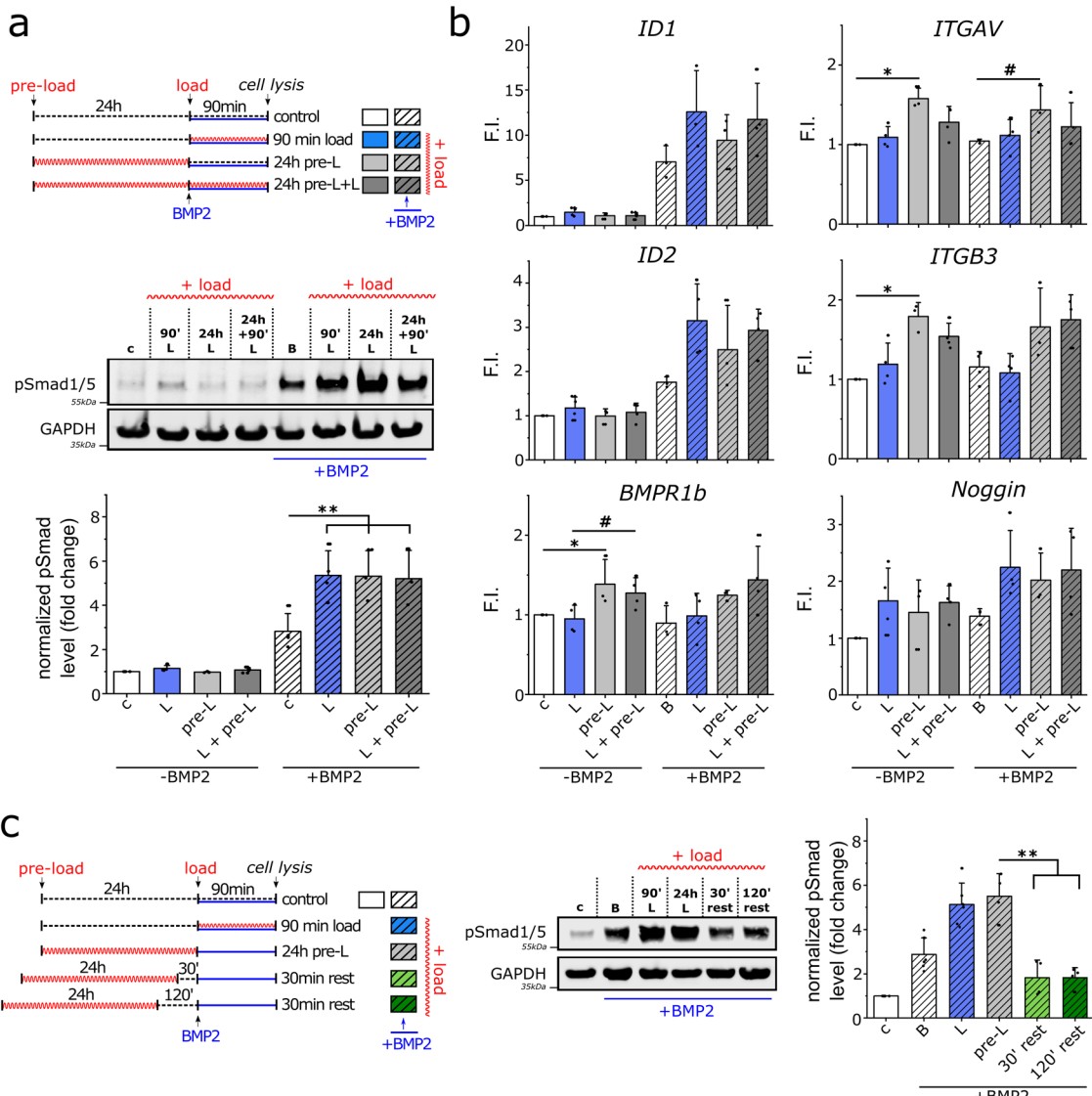

**Fig. 4 | Mechanical pre-stimulation induces a short mechanical echo but has no sustained effect on BMP signaling.** Human FOBs were seeded in collagen scaffolds, transferred into the bioreactor, and stimulated with mechanical loading (1 Hz, 10%) for 24 h or kept under non-loaded conditions. Thereafter, **a, b** cells were either immediately stimulated with BMP2 (5 nM) for 90 min with or without simultaneous mechanical loading or **c** rested for 30 or 120 min prior to BMP stimulation. Cells were lysed and **a, c** p-Smad1/5 levels were determined by western blot and **b** gene expression was analyzed by qPCR. Bar charts display means ± SD ($n \geq 3$ from at least 3 independent experiments). Statistical significance was calculated using a two-sided Mann–Whitney U test and Bonferroni correction for multiple tests (#$p < 0.1$, *$p < 0.05$, **$p < 0.01$).

Next, we elucidated the decay dynamics during BMP stimulation to find out how persistent the increase in Smad phosphorylation is once triggered by simultaneous B/L treatment. Therefore, cells were initially treated by B/L, but mechanical stimulation was stopped after 30 or 60 min, and cell lysis was performed 90 min after the addition of BMP2 (=BMP2 time interval). In addition, after 60 min of BMP2 stimulation alone, cells were mechanically stimulated for 30 min (Fig. 5a right) to investigate the responsiveness of the BMP pathway to load after initialization of the signaling cascade. Simultaneous B/L stimulation for only 30 min at the beginning or the end of the experimental setup did not increase Smad phosphorylation in cells harvested after 90 min compared to those continuously stimulated with B/L for 90 min. The 60 min B/L stimulation was able to increase the p-Smad level in the cells analyzed after 90 min compared to the BMP treatment alone, but it was lower than when stimulated simultaneously with B/L for the 90 min.

Figure 5b summarizes all loading conditions and their efficiency in terms of increasing Smad phosphorylation (=crosstalk). From this, the following conclusions can be drawn: (a) the shorter the time of mechanical pre-stimulation, the weaker the mechanical echo and thus the crosstalk between mechanotransduction pathways and BMP signaling, (b) the positive effect of cyclic compression is transient and (c) the strength and duration of crosstalk between mechanotransduction pathways and BMP signaling depend on the duration of simultaneous stimulation with BMP and cyclic compression.

## Load-induced reorganization of F-actin and focal adhesion is required to amplify BMP signaling events by short-term simultaneous mechanical stimulation

Simultaneous BMP2 and mechanical stimulation trigger an immediate early and transient amplification of BMP signaling. However, how mechanical signals are so immediately integrated into the BMP pathway is still unknown. Given this fast response, mechanical adaptation processes feeding into the BMP pathway must involve early mechanotransduction events. One of the first responses to mechanical stimulation is the reorganization of

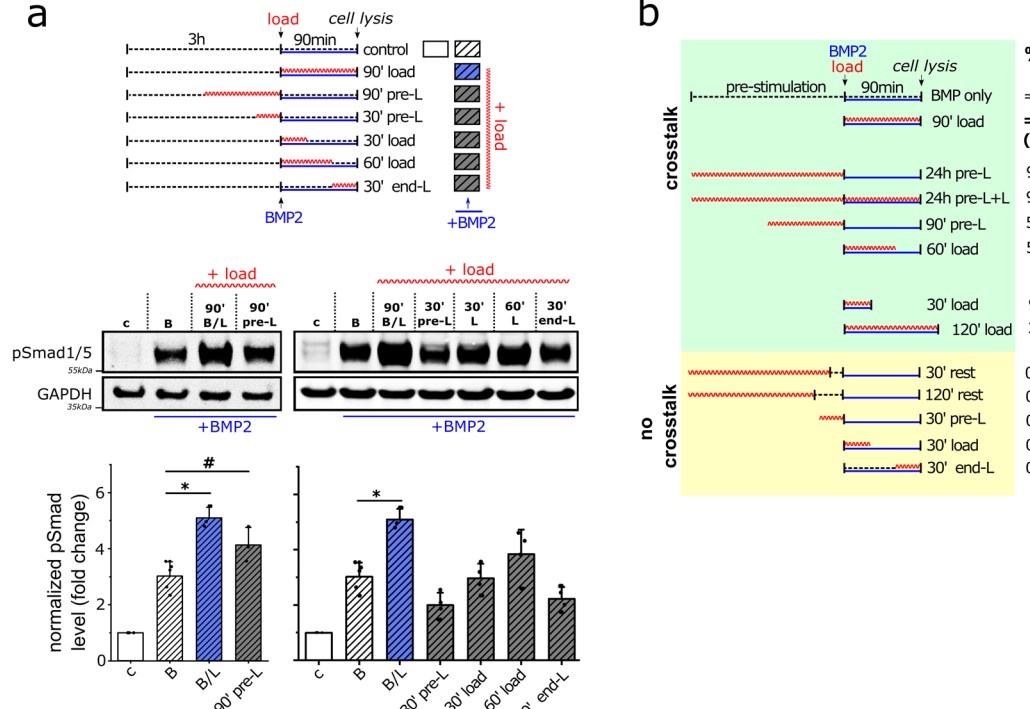

**Fig. 5 | Variations in loading conditions reveal crosstalk requirements. a** hFOBs were seeded into collagen scaffolds, transferred into the bioreactor, and stimulated with mechanical loading at different durations and timings as indicated in the schematic experimental timeline. Cells were stimulated with BMP2 (5 nM) for 90 min with or without simultaneous mechanical loading. Cells were lysed and p-Smad1/5 levels were determined by western blot. Bar charts display means ± SD

($n \geq 3$ from at least 3 independent experiments). Statistical significance was calculated using a two-sided Mann–Whitney U test and Bonferroni correction for multiple tests (#$p < 0.1$, *$p < 0.05$). **b** Summary of all loading conditions and their efficacy (in %) in potentiating Smad1/5 phosphorylation (=crosstalk). The maximum crosstalk strength (90 min of simultaneous mechanical loading (1 Hz) and BMP2 stimulation) was set to 100%, while BMP2 treatment alone was set to 0%.

the F-actin cytoskeleton together with a reinforcement of adhesion sites[38,39] which we also observed so far (Fig. 3b). Therefore, we investigated whether these adaptation processes are relevant for the positive regulation of BMP signaling by mechanical forces.

To interrupt the remolding of the F-actin cytoskeleton, the inhibitor Jasplakinolide (Jas) was used. In contrast to depolymerizing agents such as Cytochalasins[40] or Blebbistatin that result in non-physiological focal adhesion disassembly[41], Jas inhibits the depolymerization of F-actin and does not destroy the F-actin network at low concentrations, but stabilizes it[42,43]. First, the described effect of Jas on F-actin remodeling before and after Jas treatment was validated by time-lapse microscopy of the expression of LifeAct-GFP, an F-actin-binding peptide, in hFOBs (Fig. 6a). Jas treatment resulted in a drastic reduction of the fast protrusion remodeling observed for controls. Additionally, the temporal change in fluorescence signal intensity visible in the kymograph shows that actin was retracted from the outer cortex in contrast to the control. Quantification of the protrusion dynamics demonstrated that Jas treatment significantly reduced the protrusion remodeling. Since F-actin remodeling is coupled to FA reorganization[44], we speculated that Jas also inhibits FA reorganization in response to cyclic compression. To examine this, FAs were analyzed after 90 min BMP2 stimulation, 1 Hz cyclic compression, or a combination of both using either 0.1 μM Jas treatment or DMSO as a control (Fig. 6b–d).

A significant increase in the total number of FAs was induced by 90 min cyclic compression compared to the unstimulated control under DMSO treatment, while no significant increase was found under Jas treatment (Fig. 6c). The percentage of cells with medium (0.7–1 μm²) and large (>1 μm²) FAs was increased in response to cyclic compression in comparison to the unstimulated control of cells treated with DMSO (Fig. 6d). However, under Jas treatment, the percentage of cells with medium and large FAs remained the same and was therefore independent of the stimulation. The results demonstrated that the stabilization of the F-actin

cytoskeleton by Jas significantly affected load-induced FA remodeling and reinforcement. Next, we investigated whether the effects of Jas on reorganization processes interfered with the mechanical enhancement of BMP signaling. Therefore, Smad phosphorylation and ID1 expression were analyzed in hFOBs, which were incubated with 0.1 μM Jas or DMSO prior to 90 min BMP2 stimulation, 1 Hz cyclic compression, or a combination of both (Fig. 6e, f). In the DMSO-treated samples, B/L stimulation strongly and significantly increased Smad1/5/8 phosphorylation compared to BMP2 stimulation alone (Fig. 6e). In contrast, B/L stimulation under Jas treatment only slightly increased Smad phosphorylation and was significantly downregulated compared to the DMSO control. This behavior was also reflected in the expression of ID1. Whereas B/L stimulation significantly increased ID1 expression under DMSO treatment, only a slight and non-significant increase occurred under Jas treatment (Fig. 6f). Consequently, Jas treatment almost completely abolished the positive effect of cyclic compression, while basal BMP signaling remained unaffected. Together with the observed inhibition of actin remodeling and load-induced FA clustering, this led to the conclusion, that actin cytoskeletal and coupled FA adaptation responses to cyclic compression are a prerequisite for the load-induced mechano-regulation of BMP signaling.

## Discussion

Increasing evidence underlines the regulation of BMP2 signaling by biophysical cues, adding another layer to well-known biochemical regulations[14]. However, the underlying mechanisms of these regulatory processes are unclear due to a lack of systematic investigations on how different loading parameters affect the duration and strength of BMP signaling. Within this study, we investigated the impact of loading frequency, duration, and temporal relation to BMP2 supplementation on BMP signaling events. While the strength of mechanically enhanced Smad phosphorylation was saturated at the physiologically relevant frequency of 1 Hz

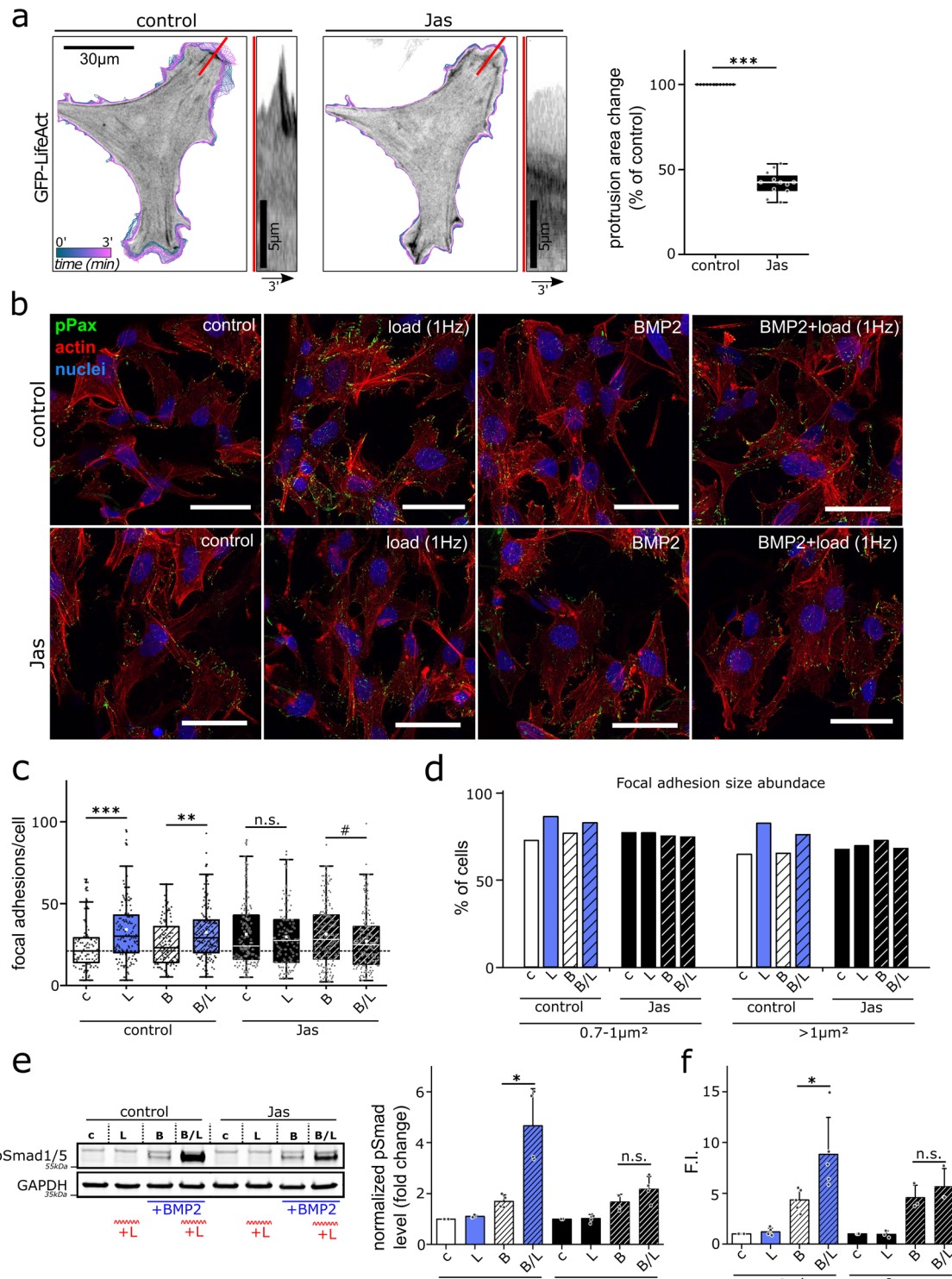

**Fig. 6 | F-actin stabilization by Jasplakinolide inhibits load-induced focal adhesion remodeling and potentiation of BMP2-induced Smad1/5/8 phosphorylation and ID1 expression. a** Representative images showing the dynamic protrusion remodeling of LifeAct-GFP expressing hFOBs during a period of three minutes before and after Jas treatment. Remodeling dynamics are visualized by overlaying all cell outlines stained from blue to pink according to the frame number. The overlay was prepared using the QuimP[61] ImageJ plugin. The kymographs taken along the red lines illustrate the different protrusion dynamics. Quantification of the mean change in protrusion area per time frame before and after Jas treatment (20–45 min after). Change in protrusion area after Jas treatment was normalized to the untreated control (=before treatment) ($n = 12$). **b** Human FOBs seeded into collagen scaffolds were incubated for 3 h in a starvation medium supplemented with

0.1 μM Jasplakinolide (Jas) or the same amount of DMSO. Subsequently, scaffolds were subjected to BMP2 stimulation (B, 5 nM), mechanical loading (L, 1 Hz, 10%), or a combination of both (B/L) for 90 min. Cells were fixated and stained for phospho-Paxillin (green), F-actin with phalloidin (red), and nuclei with DAPI (blue). Representative confocal images of stained hFOBs are depicted. Scale bar represents 50 μm. **c** Total number of phospho-Paxillin positive FA per cell and **d** percentage of cells with medium and large FAs (medium = 0.7–1 μm², large >1 μm²) were evaluated in ImageJ (in total >100 cells per condition, $n = 4$). For the analysis of **e** Smad1/5/8 phosphorylation by western blot, or **f** ID1 expression by qPCR, cells were lysed in the respective assay buffer after the bioreactor experiment ($n \geq 3$ of ≥3 independent experiments). Statistical significance was calculated using a two-sided Mann–Whitney U test (#$p < 0.1$, *$p < 0.05$, **$p < 0.01$, ***$p < 0.001$).

(occurring during locomotion), the duration of Smad phosphorylation and the enhancement of target gene expression prolonged with a further increase of the frequency towards 10 Hz (e.g. occurring during muscle contraction). This indicates that transduction of BMP signaling beyond initial Smad phosphorylation is frequency-dependent. The frequency-dependent regulation of mechanotransduction pathways is also relevant at the tissue level, as bone formation rates in rat tibiae[2] or ulna[3] increased with increasing frequency of cyclic bending (0.05–2 Hz) or compression (1–10 Hz), respectively. Interestingly, it was found that a lower compressive load (N) was required at 10 Hz than at 1 Hz to achieve the same bone formation rates[3]. Based on the similar frequency dependency observed in in vivo studies, it is therefore suggested that the crosstalk between mechanotransduction and BMP signaling is involved in the adaptation of bone to mechanical loading under homeostatic conditions.

Following the BMP pathway further downstream, we analyzed the expression of BMP receptor types that specifically bind to BMP2. Surprisingly, 24 h of cyclic compression alone increased the expression of BMP receptor 1B but not BR1A and BR2, whereas BMP2 treatment had no effect on any of the receptors. Together with the described regulation of BMP ligand expression under mechanical stimulation[23,45], this indicates that mechanical signals promote BMP signaling at multiple levels.

Furthermore, cyclic compression increased the expressions of integrin αv, β1, and β3 in a frequency-dependent manner. Thereby, the strong increase in integrin expression at 10 Hz loading frequency and the enhanced integrin clustering observed for BMP2 treated cells led to a synergistic increase in FA size and number under simultaneous stimulation (Fig. 3). Consistent with the results presented here, mechanical forces have previously been shown to induce the expression of specific integrin subtypes[46] and the growth of adhesion sites[47,48]. This suggests similar molecular mechanisms underlying the regulation of BMP signaling by active and passive biomechanical cues. A link between mechanotransduction and BMP signaling might be found in integrins, as they not only sense and transmit mechanical cues, but can also interact with BMP receptors. This interaction has been shown to modulate basal BMP signaling[32–34], but has also been suggested to mediate the regulation of BMP signaling by external mechanical forces[27]. The increase in integrin αv, β1, and β3 expression and FA assembly, together with the increased expression of BMP receptor type 1B under cyclic compression alone, was a particularly interesting observation, as both the increased amount of BMP receptors and the described BMP receptor–integrin interaction[27,32–34] could promote BMP signaling. However, the direct interaction between BMP receptors and integrins during focal adhesion remodeling in response to cyclic loading was not shown here and remains to be demonstrated in subsequent studies. Changes in the composition of the extracellular matrix can provoke alterations in the involvement of different integrin subtypes in focal adhesions potentially further influencing BMP2 signaling. Immunohistology of relevant ECM proteins, however, showed that the culture of hFOBs in the biomaterial scaffold did not alter the presence of collagen-I and fibronectin and only mildly increased the presence of collagen IV (Supplementary Data 3). Consequently, cell-secreted ECM and resulting changes in ECM composition are not expected to play a relevant role in the described BMP2 signaling response to cyclic loading.

We furthermore observed that long-term mechanical pre-stimulation induced a transient cellular adaptation process, referred to here as the mechanical echo. This adaptation resulted in increased Smad phosphorylation under BMP stimulation, even in the absence of a direct mechanical trigger, but only when BMP stimulation was applied immediately after terminating mechanical stimulation. In this situation, the duration of pre-stimulation was decisive for the effect triggered: the shorter the time of pre-loading, the weaker the effect. The phenomenon of mechanical memory has been studied almost exclusively in terms of passive biophysical cues such as substrate stiffness[37], while the role of active forces[49,50] remains unknown. While transcriptional regulations are thought to be an integral part of the mechanical memory, the early but transient induction of Smad phosphorylation upon simultaneous mechanical and BMP2 stimulation must be independent of any transcriptional regulation due to its immediacy. Therefore, mechanical adaptation processes that feed into the BMP2 pathway obviously include early mechanotransduction events, including the dynamic formation and disassembly of focal adhesions. The percentage of cells with medium (0.7–1 μm$^2$) and large (>1 μm$^2$) FAs was increased in response to cyclic compression in comparison to the unstimulated control of cells treated with DMSO (Fig. 6d). We conclude that the observed focal adhesion reorganization, which occurs within minutes of the onset[38,39] and termination of mechanical stimulation, is a marker of this process. Since we did not observe differences in the BMP signaling between cells that have been preloaded for 24 h (leading to larger focal adhesions) and non-preloaded cells (Fig. 4), we conclude that increased FAs alone are not sufficient. Only in situations where cells inefficiently adhere to a substrate resulting in artificial integrin internalization and cell rounding a reduction in cell surface integrin levels induces also a reduced BMP signaling response due to enhanced receptor internalization[36].

To verify this, the actin stabilizer Jas was used to stabilize focal adhesions and to prevent their remodeling in response to mechanical load[43] (Fig. 6). As a result, the positive effect of cyclic compression on Smad phosphorylation and ID1 expression was almost completely abolished, while basal BMP2 signaling remained unaffected (Fig. 6e, f). We thus propose that dynamic adaptations of focal adhesions in response to cyclic compression are a prerequisite for the enhancement of BMP2 signaling by mechanical forces (Fig. 7). The mechanotransduction pathway enhances BMP2 signaling at various levels by increasing the expression of BMP ligands[23,45], BMP2 receptors, and integrins, and by inducing integrin remodeling. Integrin remodeling potentially enables an increased BMP2 receptor–integrin interaction[27] to mediate the mechano-regulation of BMP2 signaling. Due to the fundamental role of this interaction, we assume that our findings are of relevance for other mechano-dependent growth factor signaling pathways, including Epidermal Growth Factor (EGF) and mammalian Target of Rapamycin Complex 1 (mTORC1), where focal adhesion involvement has previously been shown[51,52].

Together, we show that the response to mechanical loading is highly dynamic and decays rapidly when the mechanical signal is terminated, even though the cells have transcriptionally adapted to loading, e.g. by increased receptor expression. The very short mechanical echo identified here suggests that pre-stimulation is much less effective than simultaneous stimulation, which is relevant for the development of treatment strategies that aim at maximizing growth factor responsiveness. The enhancement of BMP/Smad signaling depends on load-induced F-actin remodeling and focal adhesion reinforcement. This suggests that the way how the cell is anchored to the substrate plays a crucial role in regulating BMP signaling through mechanical forces, opening up avenues for the development of therapeutic biomaterials, for example. Given the importance of BMP2 in both endogenous bone regeneration and healing of critical-sized bone defects, a comprehensive understanding may in the future help to improve treatment strategies that optimize mechanical signaling on both the organ (personalized fracture fixation systems) cellular levels (cell environments that enhance growth factor responsiveness, e.g. delivered through biomaterials). However, further in vitro and in vivo investigations are required to translate the basic findings reported here into improvements in bone healing. Beyond bone healing, the discovered role of mechanical cell adaptation in growth factor signaling might play an important role in bone development, homeostasis, and maintenance of high bone quality throughout life[53,54].

## Methods

### Cell culture

Human fetal osteoblasts (hFOBs 1.19), purchased from ATCC (Manassas, Virginia, USA), were cultured in a 1:1 mixture of Dulbecco's Modified Eagle Medium and Ham's F-12 (11320-033; Thermo Fischer Scientific, Waltham, USA) supplemented with 1 vol-% penicillin/streptomycin (A 2212; Biochrom AG, Berlin, Germany), 0.3 mg/ml Geneticin (CP11.3; Carl Roth GmbH, Karlsruhe, Germany) and 10% fetal bovine serum (FBS: S 0615; Biochrom AG, Berlin, Germany). hFOB were grown at 34 °C and 5% $CO_2$ in

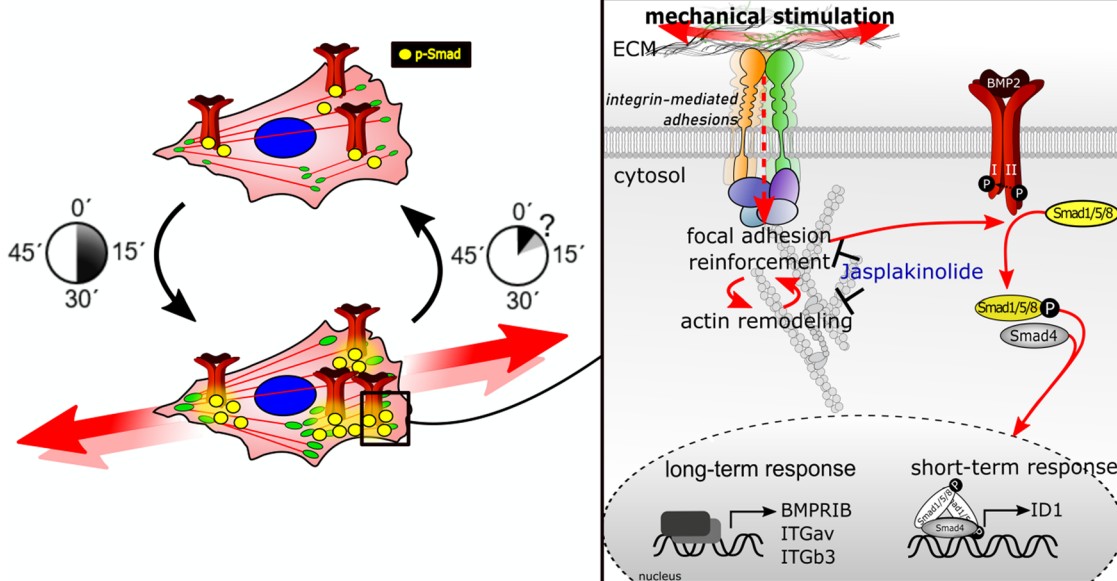

**Fig. 7 | Schematic representations summarizing the presented findings.** The potentiation of BMP/Smad signaling by concurrent BMP2 and mechanical stimulation requires load-induced focal adhesion reinforcement and actin remodeling. Long-term mechanical stimulation induces a short-lasting mechanical echo and transcriptional adaptations.

a humid incubator until 80% confluence. Primary human mesenchymal stromal cells (MSCs) isolated from bone marrow (passages 3–5) were expanded in Dulbecco's modified Eagle medium (DMEM: D5546; Sigma-Aldrich, St. Louis, USA) supplemented with 1 vol-% penicillin/streptomycin, 1 vol-% GlutaMAX (35050-038, Life Technologies, Carlsbad, USA) and 10 vol-% fetal bovine serum. Primary human dermal fibroblasts (hdF) isolated from skin biopsies (passages 5–7) were cultured in DMEM (# 41965; Gibco, Invitrogen, Loughborough, UK) supplemented with 10% fetal bovine serum, 1% penicillin/streptomycin and 1% nonessential amino acids (# K0293; Biochrom A G, Berlin, Germany). hMSCs and hDF were expanded under 37 °C and 5% $CO_2$ in a humid incubator.

**Cell seeding in collagen scaffolds**

Macroporous scaffolds from porcine collagen were utilized as a 3D cell carrier material (Optimaix®, Matricel GmbH). Cylindrical samples with a diameter of 5 mm and height of 3 mm were punched out of the raw material using biopsy punches (48501; Kai Medical). Cells were brought into suspension and a concentration of $5 \times 10^3$ cells/μl was adjusted. Scaffolds were dipped into the cell suspension and cells were allowed to adhere during a 60 min incubation at 37 °C without the addition of culture medium. Thereafter, scaffolds were washed once in a culture medium and incubated for two days under static conditions prior to all following experiments.

**Mechano-bioreactor, mechanical loading parameters, and experimental setup**

Mechanical loading experiments were conducted using a custom-made mechano-bioreactor system, previously described by Petersen et al.[55]. In brief, the system is composed of a cell culture unit and a mechanical unit. The cell culture unit consists of a bioreactor chamber, in which the mechanical stimulation takes place, a medium reservoir allowing gas exchange, and a micropump. After assembly under sterile conditions, it is combined with the mechanical unit allowing the application of defined loading patterns with online-force measurements. The system was established to mimic the mechanical conditions in a fracture gap during the initial phase of bone healing. In the bioreactor, scaffolds are mechanically stimulated by monoaxial compression, as an axial interfragmentary movement was shown to be the predominant straining regime in an in vivo osteotomy model

stabilized with an external fixator[56,57]. A compression magnitude in the range of described principal strains in the fracture region of a non-critical size sheep osteotomy was selected[58]. In detail, sinusoidal compression with a magnitude 10% of the scaffold height (=300 μm) and loading frequencies of 1 Hz, 10 Hz, and the comparably low frequency of 0.03 Hz were selected. Loading frequencies of 1 Hz and 10 Hz represent the time pattern of human locomotion[28] and muscle contraction[29], respectively.

Cell-seeded collagen scaffolds were transferred into the bioreactor system and positioned between the upper and the lower plunger. An initial pre-deformation of 50 μm was adjusted to avoid contact loss between the plunger and sample during stimulation. After sample positioning, cells were starved for 3 h prior to mechanical stimulation. All experiments were conducted under serum starvation conditions to minimize the signaling of growth factors present in the FBS. Medium containing 0% or 1% FBS was used for experiments up to 120 min or 24 h, respectively. Following starvation, cells were stimulated with 5 nM recombinant human BMP2 (E. coli origin), subjected to cyclic axial compression (varying loading frequencies and duration), or treated with a combination of both.

To investigate whether load-induced actin cytoskeleton rearrangement processes are important for the integration of mechanical signals into the BMP pathway, the actin-binding macrocyclic peptide Jasplakinolide (Jas, J4580, Sigma-Aldrich), an inhibitor of actin depolymerization[42], was used as an agent to stabilize actin filaments. Jas at a concentration of 0.1 μM, or the same amount of the solvent dimethyl sulfoxide (DMSO, 276855, Sigma-Aldrich) was supplemented to the culture medium at the beginning of the starvation phase. After three hours of starvation, scaffolds were subjected to cyclic mechanical compression of 10% with a frequency of 1 Hz for 90 min. Thereafter, cells were either fixed in 4% PFA for subsequent immunofluorescence staining or lysed in the respective assay buffer for western blot or qPCR analysis.

**Cell lysis and western blot analysis**

Cells were lysed in RIPA buffer (89900, Thermo Fischer Scientific) containing phosphatase (PhosSTOP™, Merck KGaA) and protease inhibitors (cOmplete™ Protease Inhibitor Cocktail, Merck KGaA). Samples were incubated for 4 min in the lysis buffer on ice, vortexed, sonicated for 30 s, and vortexed again. Subsequently, scaffolds were centrifuged through a tip of a 0.5–10 μl pipette, thereby collecting the lysate while the scaffold

remained dry inside the tip. Lysates were frozen at −20 °C until further western blot analysis.

Protein lysates were mixed with the 4× loading buffer (928-40004, LI-COR Biosciences), loaded into 4–12% bis-tris polyacrylamide gels (NP0336Box, Thermo Fischer Scientific) and SDS-PAGE was conducted. Thereafter, proteins were transferred on nitrocellulose membranes (741280, Macherey-Nagel GmbH & Co. KG) by western blot. Membranes were blocked for 1 h in TBS containing 5% IgG-free albumin (3737.3, Carl Roth GmbH) and incubated in the respective primary antibody overnight at 4 °C following the manufacturer's instruction. After washing with TBS-T (0.1% tween), membranes were incubated for 2 h with the secondary antibodies conjugated to fluorescence tags (P/N 925-32211, Li-Cor Biosciences). Proteins were visualized and quantified using the Odyssey Infrared Imager (Li-Cor Biosciences) and the corresponding software. GAPDH served as a loading control to which the proteins of interest were normalized. The following antibodies were used: glycer-aldehyde-3-phosphate dehydrogenase (GAPDH; #2118, Cell Signaling Tech.), phosphorylated Smad1/5/8 (#9511, Cell SignalingTech.), phosphorylated Smad1/5 (#9516, Cell Signaling Tech.).

### RNA isolation, reverse transcription, and quantitative real-time PCR

Total RNA was isolated using the PureLink® RNA Mini Kit (12183018A, Life Technologies) and DNA digestion was performed using On-column PureLink® DNase (12185-010, Invitrogen). Reverse transcription of 500 ng RNA to cDNA was conducted using the iScript™ cDNA Synthesis Kit (#170-8891, BIO-RAD). SYBR green-based quantitative real-time PCR (qPCR) was carried out and mean normalized gene expression was calculated using the efficiency corrected ∆∆CT–method with HPRT as a housekeeping gene. Primer sequences are provided in Supplementary Data 6.

### Immunofluorescence staining and imaging

Immediately after the loading experiments, cells were fixed for 2 h in 4% paraformaldehyde, quenched for 1 h in 50 mM ammonium chloride, and subsequently incubated for 1 h in 5% gelatin solution at 37 °C. Thereafter, gelatin was solidified at 4 °C to stabilize the scaffold structure during the cutting along the symmetry axis of the scaffold. After cutting, gelatin was washed out by incubation in PBS at 37 °C. Integrin-mediated adhesion sites were visualized using phosphor-Paxillin (Y118) antibody (#2541, Cell Signaling Tech.) The F-actin cytoskeleton was stained using Phalloidin-Alexa 488 (# A12379, Life Technologies) and cell nuclei were stained with 4′-6-diamidino-2-phenylindiole (DAPI, # D1306, Life Technologies). Extracellular matrix proteins (Supplementary Data 3) were stained using anti-fibronectin antibody ab23750, anti-collagen-I antibody ab138492 [EPR7785], and anti-collagen-IV antibody ab6586, all from Abcam. Vinculin (Sigma-Aldrich, V9131) and integrin αV (Santa Cruz, sc-9969) (Supplementary Data 4) were stained using antibodies as indicated by the manufacturer. The collagen scaffold was visualized by second harmonic generation (SHG) microscopy, a label-free method to image fibrillar collagen. Images were taken using a Leica TCS SP5II confocal microscope at 63× magnification.

### Luciferase reporter gene assay

Luciferase reporter gene assay using C2C12 stably transfected with a BMP response element (BRE)-luciferase reporter was carried out according to the protocol from Herrera and Inman[59]. In brief, C2C12 BRE-luc reporter cells were seeded at a density of $1.3 \times 10^4$ cells/cm² in a 24-well plate with DMEM (low glucose, D5546, Sigma-Aldrich) supplemented with 10 vol.% Fetal Bovine Serum, 1 vol.% GlutaMAX™ (35050-038, Thermo Fischer) and 1 vol.% Geneticin and cultured overnight at 37 °C with 5% CO₂ in a humidified incubator. Thereafter, cells were washed once with PBS (500 μl/well) and starved for 8 h in 300 μl/well DMEM (low glucose) supplemented with 1% GlutaMAX™. For the BMP2 concentration-dependent standard curve, 50 μl of defined BMP2 concentrations were added to the cells. For the test samples, the starvation medium was exchanged with the collected

bioreactor culture medium (350 μl/well) and incubated for a further 16 h. Subsequently, cells were washed once with PBS, lysed using 100 μl of 1× lysis buffer (Pierce™ Firefly Luciferase Glow Assay Kit, 16176, Thermo Fisher), and agitated for 15 min at 400 rpm on top of a thermo-mixer. Of each sample, 20 μl were transferred to a 96-well plate and 50 μl of the firefly luciferase substrate was added. After 10 min incubation, a luminescent signal was measured using a microplate reader (Tecan Infinite pro-2000). The luminescence signal was normalized to the total protein content, which was measured in parallel using the BCA assay according to the manufacturers' instruction (Pierce™ BCA Protein Assay Kit, 23225, Thermo Fisher).

### Lentiviral transduction of hFOBs with LifeAct-GFP

To visualize and follow F-actin remodeling in living cells, the GFP-tagged F-actin-binding peptide LifeAct (60141, Ibidi) was introduced into hFOBs by viral transduction. In contrast to GFP-actin or other actin labeling methods, LifeAct was not found to interfere with the actin dynamics[60]. hFOBs were seeded at 50% confluence in 6 well plates and incubated overnight at 5% CO₂ and 37 °C. Thereafter, the medium was removed and a transduction medium, containing no antibiotics, 10% heat-inactivated FCS, and 8 μg/ml hexadimethrine bromide (Polybrene), was added. The lentiviral vector was supplemented to the cells with a multiplicity of infection of 2. After 20 h of incubation, the medium containing the lentiviral particles was removed and an expansion medium was added. Five days after transduction, cells were transferred into a 25 cm² cell culture flask incubated overnight and subjected to positive selection using 1 μg/ml puromycin until only transduced cells remained. hFOBs stable expressing LifeAct-GFP are called hFOB-LA in this study.

### Live-cell-imaging and analysis

Time-lapse live-cell imaging was performed to investigate F-actin reorganization processes before and during Jas treatment (0.1 μM). To maintain cell culture conditions (37 °C, 5% CO₂) during the experiment, the Leica TCS SP5 confocal microscope was used in combination with an incubation chamber and a gassing unit. hFOBs expressing LifeAct were seeded in 8-well chamber slides (80826, Ibidi) and image stacks of 5 μm with a z-spacing of 1 μm were recorded every 6 s over 3 min using the 63× objective at 1024 × 1024 pixel resolution. Image sequences of the same cell were recorded before and after 20–45 min Jas treatment. Protrusion dynamics of whole cell protrusions were analyzed using a custom-made Image macro. In brief, z-stacks were projected, cell outlines were contoured for each time point and the area in between two consecutive ROIs was determined. Mean area change per time frame was calculated for all image sequences and the mean area change per time frame after Jas treatment was normalized to the mean area change before treatment. Representative images showing the cell outline change over time were prepared using the QuimP[61] ImageJ plugin.

### Statistics and reproducibility

Statistical significance between groups was assessed by the non-parametric, two-sided Mann–Whitney test with Bonferroni correction. P-values below 0.05 were considered significant ($^{\#}p < 0.1$, $^*p \le 0.05$, $^{**}p \le 0.01$, $^{***}p \le 0.001$). OriginPro 2015G by OriginLab Corporation was used for statistical analyses and to prepare all charts.

### Reporting summary

Further information on research design is available in the Nature Portfolio Reporting Summary linked to this article.

### Data availability

Raw western blot scans of blot data shown in the main figures are provided in the supplementary data file (Supplementary Data 7–10). Primary source data of the presented plots are summarized in a supplementary data source file (Supplementary Data 11). Additional raw data are available upon reasonable request.

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

## Acknowledgements

The authors acknowledge Hans Leemhuis (Matricel GmbH, Herzogenrath, Germany) for providing collagen scaffolds. Furthermore, we like to thank Dr. Maria Reichenbach for their cooperation and helpful discussions. S.G. and E.B. were supported by the DFG Graduate School 203 "Berlin-Brandenburg School for Regenerative Therapies". This study was supported by the German Research Foundation (DFG) through funding for the Collaborative Research Center 1444, sub-project P03.

## Author contributions

S.G., P.K., and A.P. designed the study. S.G., E.B., R.G., and A.P. carried out the experiments. S.G. and E.B. analyzed the experiments. S.G. and E.B. together with A.P., G.D., and P.K. interpreted the data. S.G., E.B., and A.P. wrote the manuscript. All authors reviewed the manuscript and approved the final version.

## Funding

## Competing interests

The authors declare no competing interests.
