## [Peer Review file · Communications Biology]

Temporal regulation of BMP2 growth factor signaling in response to mechanical loading is linked to cytoskeletal and focal adhesion remodeling

Corresponding Author: Professor Ansgar Petersen

Figures originally included in the author's rebuttal have been redacted from this file.

Version 0:

Reviewer comments:

Reviewer #1

(Remarks to the Author)

In this study, Gorlitz et al. explore the effect of loading frequency and duration on BMP signalling of hFOB cells within a 3D mimicking microenvironment. The rigorous investigation offers valuable insights into the mechanism governing BMP signalling. The manuscript is well presented and could be beneficial to researchers studying bone regeneration. I believe this study is acceptable for publication in *Communication Biology* following minor revisions to enhance its overall quality. Line 48: Sentence does not seem correct. Line 217 refers to Fig 53B.

Reviewer #2

(Remarks to the Author)

The authors of this manuscript report the role of mechanical cues in enhancing the signalling pathway of bone development and regeneration growth factor BMP-2 (Bone morphogenetic protein-2). They established a mechano-bioreactor system mimicking the in-vivo environment where osteoblasts were allowed to adhere to a 3D cell carrier material. The cells underwent BMP-2 stimulation and cyclic compression at different frequencies and durations. The authors demonstrated that intensity and duration of BMP signalling is directly proportional with increasing load frequency synchronised with focal adhesion size and number. The findings on long-term mechanical stimulation showed surge in expression levels of BMP receptor type 1B, specific integrins and integrin clustering. This also resulted in a mechanical echo that enhanced BMP signalling, even when the cyclic compression was terminated. They concluded through drug tests that load-induced focal adhesion and cytoskeletal remodelling is essential to amplify the BMP signalling by cyclic compression and generation of mechanical echo.

The authors have provided a well-designed and carefully executed experimental setup to demonstrate an interesting take on the BMP-2 signalling pathway connecting it to the biophysical cues that shape the microenvironment of the osteoblasts. The data analysis was thorough and rigorous, making this a solid paper.

Here are a few minor suggested revisions listed below to further strengthen the manuscript:

1. It would be helpful to explain the full form of BMP-2 and its significance at the beginning of the abstract.
2. Here the authors have provided the expression levels of integrins during differential loading and compression+BMP2 stimulation (B/L) (Figure 2F). It would be helpful if the authors can provide fixed sample images showing the distribution of integrins and their clustering at the focal adhesion. This can give a robust insight into integrin clustering, focal adhesion size and actin remodelling.
3. The manuscript is well written generally, but there are a few typos that should be corrected. For example, please rephrase the sentence (line 47-48) "Bone Morphogenetic Protein 2 (BMP-2) is known to play a central role in bone development and regeneration and has gained Besides its indispensable...". Some sentences in the manuscript can also be revisited for easier understanding. For example, the sentence "We wanted to further understand how BMP signalling responds when the

time of pre-stimulation is shortened so that cells do not have time to fully adapt to the altered mechanics at all levels, and thereby find out which cellular adaptation processes are responsible for the mechanical echo and the subsequent rapid decay of the crosstalk between mechano-transduction and BMP signalling” can be simplified to gain clarity.

Reviewer #3

(Remarks to the Author)

The manuscript by Gorlitz et al describes the aspects of the BMP signalling pathway in the context of how cells in a 3D mechanical bioreactor system respond to a designed temporal stimulation of BMP2 and mechanical compression. The mechano-bioreactor design the authors use allows them to bridge the gap between in vivo bone regeneration and in vitro parameter control. In particular the temporal control of compression and BMP addition. Changes in transcript expression levels of a host of proteins that are (in)directly involved in the BMP pathway were measured as well as the first downstream effect in the BMP pathway: SMAD phosphorylation.

Measuring the direct downstream effect of BMP signalling (phosphorylation of SMAD) the authors find that mechanical stimulation augments BMP signalling. This effect is rather immediate (Fig 2A-D, and earlier studies from the authors) and long-term mechanical stimulation also increases negative regulators downstream of SMAD (Fig 4C). These data seem to suggest that the feedforward memory for BMP stimulation is much shorter-lived than feedback memory of the system. To understand the relation between dynamic F-actin remodelling and the augmented BMP signalling upon mechanical stimulation the authors stabilize the actin using Jasplakinolide. There was a complete loss the SMAD phosphorylation increase nor was there a significant increase in the downstream ID1 transcript expression level when BMP signalling was combined with mechanical stimulation in the perturbed situation (Fig 6E).

In general the manuscript is well written but as a whole it is a bit confusing and appears to consist out of two sets of data. One displaying a message of the temporal (dis-)entanglement of the mechanical stimulus and BMP2 signalling (Fig 2, 4 and 5) and a second discussing focal adhesions (Fig 3 and 6). Their relation is not entirely clear. Focal adhesions are as expected bigger in situations where forces are applied to cells (one of many examples is Riveline J Cell Biol 2001). Disrupting actin turnover using Jas directly affects loading-related focal adhesion growth (Galbraith Science 2007, Parsons Nat Rev Mol Cell Biol 2010). Even though the authors subsequently show that Jas perturbation also disrupts loading-related downstream BMP signalling this does not warrant a link with focal adhesion maturation (as mentioned in the title). Perhaps the authors in this regards could make use of the benefits of their system: they could analyse/correlate single cell SMAD nuclear localisation (or phosphorylation) with number/size distribution of focal adhesions.

Some more specific comments:

1-The authors find that co-stimulation of BMP and cyclic compression enhances BMP signalling. Using 3 frequencies they find a frequency dependence: 0.03 Hz showed an increased p-SMAD in the first 30 min, 1 Hz an increased p-SMAD up to 90 min whereas mechanical stimulation at 10 Hz the effect on p-SMAD is seen up to 120 min (Fig 2 A-D). Can the authors comment on the temporality of the stimulation effect? Why do the p-SMAD levels come down at different timepoints depending on the frequency? If this is a negative feedback it must be very different from the negative feedback acting further downstream as assessed from Fig 4C.

2-In the same set of experiments it is not clear how the ID1 and ID2 transcript upregulation is related to the levels of upstream p-SMAP. While the 0.03Hz stimulation does not significantly enhance SMAD phosphorylation it does give an 8-fold or 3-fold increase for ID1 and ID2 transcripts, respectively. The latter effect does seem to suggest a strong BMP response and it is not clear why 0.03Hz is not taken forward for the 24h experiments.

3-Later on the occurrence of a negative feedback initiated by the long term mechanical stimulation needs to be invoked to explain the lack of a direct correlation in SMAD phosphorylation and ID1/2 transcript expression (Fig 4A and B). This is rather vague. Is there any experimental/literature evidence of what that axis might be? Does the upregulation of those negative downstream regulators also explain the lower-fold transcript levels in ID1/2 in Fig 2F (compared to Fig 2E: 90 min versus 24hrs, respectively)?

4-The major perturbant the authors use is acting directly on actin turnover. Instead of measuring the focal adhesions that could function as a measure of cell anchorage (FA number) and internal tension (FA number and FA size) the authors should perhaps focus on the total F-actin content. Alternatively other perturbants can be used, such as the Myosin 2 inhibitor blebbistatin (Pasapera J Cell Biol 2010).

5-Collagen is used as the extracellular matrix on the 3D scaffold. While this is expected to stimulate a subset of integrins there is a change in integrin transcript levels that could be indicative of a changing extracellular matrix (Fig 2F and 4B). In particular there are increases in the transcript levels of alpha_v, beta₁, and beta₃ indicative of RGD containing ECM (Hynes Cell 2002). Would a change in ECM chemistry change the BMP signalling response? Could this explain the difference summarized in Fig 5, for example between [24h L + 90° B; 98% crosstalk] and [90° L pre B; 50% crosstalk]?

6-Is the memory (mechanical echo) frequency dependent?

Minor comments:

7-Line 47-50: Seems to require rephrasing.

8-Line 217: Figure reference is likely to 2F.

Author Rebuttal letter:

Response to the reviewers's comments

We have re-worked the manuscript to address the reviewers's concerns and we are convinced of the
improved thoroughness and consistency of our data and the derived conclusions. We have highlighted
all changes and amendments in the following point-by-point response.

1 Reviewer #1

o Comment 1

In this study, Gorlitz et al. explore the effect of loading frequency and duration on BMP signalling of
hFOB cells within a 3D mimicking microenvironment. The rigorous investigation offers valuable insights
into the mechanism governing BMP signalling. The manuscript is well presented and could be beneficial
to researchers studying bone regeneration. I believe this study is acceptable for publication in
Communication Biology following minor revisions to enhance its overall quality.

Line 48: Sentence does not seem correct.

Line 217 refers to Fig 53B.

Response

We thank the reviewer for the positive response. We have corrected the typos accordingly.

2 Reviewer #2

o Comment 1

It would be helpful to explain the full form of BMP-2 and its significance at the beginning of the abstract.

Response

We would like to thank the reviewer for the constructive feedback concerning our manuscript. We have
revised the abstract to address the reviewer's comment.

o Comment 2

Here the authors have provided the expression levels of integrins during differential loading and
compression+BMP2 stimulation (B/L) (Figure 2F). It would be helpful if the authors can provide fixed
sample images showing the distribution of integrins and their clustering at the focal adhesion. This can
give a robust insight into integrin clustering, focal adhesion size and actin remodeling.

Response

We agree with the reviewer that the choice of phospho-paxillin for analyzing focal adhesion adaptation
to load should be better explained. As paxillin is an intracellular adaptor protein localizing to focal
adhesions which, similar to vinculin, it directly interacts with integrins. Phosphorylation of paxillin by
focal adhesion kinase provides additional docking sites for further adapter proteins such as Src 1. We
therefore chose phospho-paxillin as a marker of maturing focal adhesions which hold greater relevance
for cell anchoring, integrin signaling and cytoskeletal tensioning. We have included additional histology
(new Additional file 4) that illustrates the co-localization either of vinculin and phospho-paxillin or of
alpha-V Integrin and phospho-paxillin. These data underline that particularly large focal adhesions as a
focus of this work are enriched in any of these markers without a specific preference. Yet, phospho-
paxillin provides a greater signal specificity due to the detection of the phosphorylated version. We have
added the following sentence to motivate the use of phospho-paxillin for the visualization of focal
adhesions:

"Phospho-Paxillin represents a more selective marker of focal adhesion maturation compared to other
markers such as integrins and vinculin. Yet, a strong co-localization of either of these markers with
phospho-paxillin was observed, indicating that phospho-paxillin positive FAs are generally enriched in
integrins and vinculin." (lines 174-177)

o Comment 3

The manuscript is well written generally, but there are a few typos that should be corrected. For
example, please rephrase the sentence (line 47-48) "Bone Morphogenetic Protein 2 (BMP-2) is known
to play a central role in bone development and regeneration and has gained Besides its
indispensable". Some sentences in the manuscript can also be revisited for easier understanding. For
example, the sentence "We wanted to further understand how BMP signalling responds when the time
of pre-stimulation is shortened so that cells do not have time to fully adapt to the altered mechanics at
all levels, and thereby find out which cellular adaptation processes are responsible for the mechanical
echo and the subsequent rapid decay of the crosstalk between mechano-transduction and BMP

signalling can be simplified to gain clarity.

Response

We have re-worked the manuscript and the particular passages indicated by the reviewer to increase
clarity.

3 Reviewer #3

o Comment 1

In general the manuscript is well written but as a whole it is a bit confusing and appears to consist out
of two sets of data. One displaying a message of the temporal (dis-)entanglement of the mechanical
stimulus and BMP2 signalling (Fig 2, 4 and 5) and a second discussing focal adhesions (Fig 3 and 6).
Their relation is not entirely clear. Focal adhesions are as expected bigger in situations where forces are
applied to cells (one of many examples is Riveline J Cell Biol 2001). Disrupting actin turnover using Jas
directly affects loading-related focal adhesion growth (Galbraith Science 2007, Parsons Nat Rev Mol
Cell Biol 2010). Even though the authors subsequently show that Jas perturbation also disrupts loading-
related downstream BMP signalling this does not warrant a link with focal adhesion maturation (as
mentioned in the title). Perhaps the authors in this regards could make use of the benefits of their
system: they could analyse/correlate single cell SMAD nuclear localisation (or phosphorylation) with
number/size distribution of focal adhesions.

Response

We appreciate the reviewers' suggestions and concerns that helped us to improve the manuscript.
Despite the effect of the inhibitor on the actin cytoskeleton, the primary focus of this work was to exploit
the influence of Jasplakinolide treatment on focal adhesion turnover - an effect well documented and
indicated by the reviewer. The key effect of the inhibitor is the prevention of a dynamic cellular adaption
in response to load. Due to the strong entanglement of cytoskeletal networks and cell adhesion clusters,
the provoked alterations in the cytoskeleton also effect the focal adhesions through which cytoskeletal
forces are anchored to the extracellular matrix. As inhibitors acting on the cytoskeleton are well
established and their mechanisms of action are known, we used Jasplakinolide in this study to alter focal
adhesion remodeling and adaptation to load. We have emphasized this point in the manuscript (e.g. lines
300-302, 475-476)

Concerning the very reasonable suggestion to correlate single cell Smad nuclear location to focal
adhesion size, we would like to clarify that even though we attribute enhanced BMP2 signaling to focal
adhesion maturation, this does not imply that the extend of Smad phosphorylation correlates with focal
adhesion size on a single cell level. Our existing data underline this as we did not observe visible
differences in the phospho-Smad levels of cells pre-loaded for 24h (which exhibit enlarged FAs) and
non pre-stimulated cells (which show smaller and lesser FAs) (Figure 3, comparison of \hat{c} and \hat{L}
1Hz). From these data we conclude that the key element is a dynamic adaptation process of focal
adhesions in response to mechanical triggers (here cyclic axial compression), not the size or maturation
status of focal adhesions per se. The degree by which this focal adhesion adaptation takes place scales
with the loading intensity. Hence the time window in which a beneficial effect on Smad signaling can
be induced also depends on the frequency with a rather transient effect at low frequencies (1Hz) and a
more persistent one at higher frequencies (10Hz) (Figure 2). Furthermore, cellular adaptation to cyclic
loading cannot be expected to take place only temporally when mechanical loading is switched on, but
is a continuous process of adapting to an environment featuring not only temporal alterations but also
the spatial heterogeneity of cyclic straining as visible from Fig. 1C.

In parallel to this manuscript, we have submitted findings about the influence of substrate stiffness on
BMP signaling, which is currently in revision at Biomaterials. In this study, we do not observe
differences in BMP signaling on PDMS substrates, despite an increased FA size with increasing
substrate stiffness. Yet, a very soft version of the collagen scaffold used there that permits local material
deformation by cell forces in an oscillating manner, BMP signaling was also upregulated. The
deformation induced by the migrating cells led to a heterogenous, non-linear deformation comparable
to the material deformation resulting from the external mechanical loading applied in this study (Figure
1). Hence, we see similarities between the two situations as in both cases a dynamic environment is
created in which cells, through migration, oscillate between strained and non(less)-strained regions. We
believe that the dynamic iteration between differently strained regions resembles a key element that
provokes a continuous cellular adaptation through FA assembly.

We have included these considerations into the manuscript at the respective passages, particularly in the
discussion (lines 458-467). To avoid misunderstanding, we have further adapted the manuscript title to
indicate "focal adhesion remodeling" instead of "focal adhesion maturation".

o Comment 2

The authors find that co-stimulation of BMP and cyclic compression enhances BMP signalling. Using 3
frequencies they find a frequency dependence: 0.03 Hz showed an increased p-SMAD in the first 30 min,
1 Hz an increased p-SMAD up to 90 min whereas mechanical stimulation at 10 Hz the effect on p-SMAD
is seen up to 120 min (Fig 2 A-D). Can the authors comment on the temporality of the stimulation effect?
Why do the p-SMAD levels come down at different timepoints depending on the frequency? If this is a

negative feedback it must be very different from the negative feedback acting further downstream as
assessed from Fig 4C.

Response

With this work, we provide evidence that the dynamic adaptation of focal adhesions is a prerequisite for
the observed synergism in BMP signaling â most likely resulting from the direct interaction of BMP
receptors and integrins^{2,3}. In this context, the increase in focal adhesion maturation was frequency
dependent (Figure 3). As the rate at which new focal adhesions can be formed is likely to be limited, the
time until cells adopt a new steady state (and when the beneficial effect of concomitant compared to
BMP-only stimulation is lost), scales with the frequency. Based on our data for Smad phosphorylation
this time window is in the range of 30-90 minutes (low frequency) and up to 120 minutes (high
frequency). In an attempt to achieve the original steady state (unloaded), the removal of mechanical
loading leads to an internalization and retraction of focal adhesions during which BMP receptors have
been observed to become internalized as well³. This explains why the removal of mechanical loading
either shortly before or at the moment of BMP stimulation resulted in a less pronounced or even lacking
synergism (Figure 5). All these processes are distinct from classical negative feedback loops inherent to
the BMP signaling cascade such as the BMP-induced expression of extracellular antagonists (Noggin,
Figure 2F) and inhibitory Smads (Smad-7, Figure 2F). Neither during concomitant stimulation, nor long
term pre-stimulation these inhibitory regulators are likely to be relevant as any increase in their
expression was only observed for 10Hz frequency and after 24h and can only be interpreted as the result
of the significantly enhanced induction of signaling in the first place. We included these considerations
in the text (see lines 213-215).

o Comment 3

In the same set of experiments, it is not clear how the ID1 and ID2 transcript upregulation is related to
the levels of upstream p-SMAP. While the 0.03Hz stimulation does not significantly enhance SMAD
phosphorylation it does give an 8-fold or 3-fold increase for ID1 and ID2 transcripts, respectively. The
latter effect does seem to suggest a strong BMP response and it is not clear why 0.03Hz is not taken
forward for the 24h experiments.

Response

The regulation of gene expression by Smad proteins requires their nuclear translocation upon
phosphorylation. We have recently shown that mechanical loading enhances the nuclear translocation
of phosphorylated Smad proteins⁴. In the same study, we observed an enhanced Smad phosphorylation
as early as 15 minutes after concomitant BMP stimulation and mechanical loading. Hence, an increased
signaling might occur already earlier. While levels of phosphorylated Smads are only detected for the
time point of analysis, gene transcript levels reflect the sum of transcription factor activity over a distinct
time window where tiny differences in transcription factor activity (=Smad phosphorylation) can
accumulate into more pronounced differences in the downstream signaling (gene expression).
In general, our data suggest that the level of signal amplification through the concomitant stimulation
scales with the intensity (=frequency) of mechanical loading. After 24h, we no longer observed an
increased expression of ID1 and ID2 between BMP stimulation and concomitant stimulation (BMP +
load) for 1Hz frequency (Figure 2F). The same observation can be made for focal adhesion count and
size (Figure 3), where only 10Hz frequency resulted in a visible difference. Together this suggests that,
despite the beneficial effect on signal initiation, it requires a physiologically meaningful loading
frequency to provoke a persistent cellular response. Due to this lack of relevance and significance, we
excluded 0.03Hz from further experiments. We have included these consideration in the main text of
the manuscript (see lines 140-142)

o Comment 4

Later on the occurrence of a negative feedback initiated by the long term mechanical stimulation needs
to be invoked to explain the lack of a direct correlation in SMAD phosphorylation and ID1/2 transcript
expression (Fig 4A and B). This is rather vague. Is there any experimental/literature evidence of what
that axis might be? Does the upregulation of those negative downstream regulators also explain the
lower-fold transcript levels in ID1/2 in Fig 2F (compared to Fig 2E: 90 min versus 24hrs, respectively)?

Response

We would like to refer to a previous response (see comment 3) in which we argued that the observed
transcript levels of a gene of interest can be regarded as the sum of transcription factor activity over a
distinct time frame. In this regard, a lack of a direct correlation between phospho-Smad and target gene
expression levels is not sufficient to speculate on active negative feedback mechanisms being initiated.
We therefore do not believe that the observed differences are due to an active negative feedback
mechanism. However, our data (Supplementary Figure 2) show that with 24h of mechanical loading,
inhibitory Smad (Smad7) and Smad Ubiquitin Ligase (Smurf1/2) become upregulated which potentially
interfere with cellular BMP signaling responses. However, for 1Hz frequency, these differences are on
an overall low level and more relevant for the 10Hz frequency which we did not use for long term pre-
loading of cells.

o Comment 5

The major perturbant the authors use is acting directly on actin turnover. Instead of measuring the

focal adhesions that could function as a measure of cell anchorage (FA number) and internal tension
(FA number and FA size) the authors should perhaps focus on the total F-actin content. Alternatively
other perturbants can be used, such as the Myosin 2 inhibitor blebbistatin (Pasapera J Cell Biol 2010).

Response

We appreciate this intriguing thought. Due to the direct coupling of the actin cytoskeleton to focal
adhesions it is practically impossible to selectively interfere with actin turnover but not focal adhesion
formation and vice versa. As suggested, we have performed additional experiments using the cell force
inhibitor blebbistatin that led to a strong reduction in focal adhesion count and size. Furthermore, this
treatment not only reduced the basal BMP signaling, but also prevented any load-induced synergism.
Together, these new data underline our conclusion that the increase in the BMP signaling response is
linked to a dynamic focal adhesion remodeling and maturation process. Although acting primarily on
the cytoskeleton, the advantage of Jasplakinolide is that it does not directly lead to a disassembly of the
FAs as observed for blebbistatin. Both inhibitors hereby interfere with the load-induced synergism
which can be linked to the potential to form and remodel focal adhesions. The fact that by using
blebbistatin focal adhesion size decreases to a non-physiological level is in line with previous reports in
which a non-physiological cell-rounding due to inefficient adhesion on very soft substrates (<1kPa)
leads to integrin internalization and ultimately reduced BMP signaling³. We have added this
consideration into the discussion part of the main manuscript (see lines 409-412).

o Comment 6

Collagen is used as the extracellular matrix on the 3D scaffold. While this is expected to stimulate a
subset of integrins there is a change in integrin transcript levels that could be indicative of a changing
extracellular matrix (Fig 2F and 4B). In particular there are increases in the transcript levels of alpha_v,
beta₁, and beta₃ indicative of RGD containing ECM (Hynes Cell 2002). Would a change in ECM
chemistry change the BMP signalling response? Could this explain the difference summarized in Fig 5,
for example between [24h L + 90% B; 98% crosstalk] and [90% L pre B; 50% crosstalk]?

Response

We agree with the reviewer that the question of how cells anchor within the here used biomaterial niche
is an important point – particularly when focusing on integrins. We have included additional histology
that demonstrates the adsorption of fibronectin on the collagen walls of the biomaterial scaffold. This
fibronectin originates from the bovine serum which is added as a supplement to the cell culture medium
(new Additional file 3) and forms a thin layer directly during cell seeding. In contrast to the culture of
primary human fibroblasts in the scaffold⁵, this layer is mostly unchanged by the cell line used in this
study throughout the pre-incubation time (2 days prior to stimulation). Particularly, no fibronectin fibers
were found in the scaffold as they are typically deposited by fibroblasts⁵ or MSCs. We furthermore
stained selectively for human collagen type I and type IV as important fibrillar respectively network-
forming ECM proteins. Together, these new data indicate that the hFOBs used in this study are adhering
to a mixture of the original collagen-I material, a thin adsorbed fibronectin layer which remains mostly
stable throughout culture and a minor deposition of collagen type IV by hFOBs over 24h of culture. We
therefore assume that changes in recognized adhesion patterns are negligible and cannot explain
differences in BMP2 signaling between conditions representing different culture durations.
We have added the following passage to the main text (lines 413-419):

–Changes in the composition of the extracellular matrix can provoke alterations in the involvement of
different integrin subtypes in focal adhesions potentially further influencing BMP-2 signaling.
Immunohistology of relevant ECM proteins however showed that the culture of hFOBs in the biomaterial
scaffold did not alter the presence of collagen-I and fibronectin and only mildly increased the presence
of collagen IV (Additional File 3). Consequently, cell-secreted ECM and resulting changes in ECM
composition are not expected to play a relevant role in the described BMP-2 signaling response to cyclic
loading.â

o Comment 7

Is the memory (mechanical echo) frequency dependent?

Response

With this study, we provide evidence that a dynamic adaptation process of focal adhesions is linked to
an increased BMP signaling response. As our data indicate that the degree by which this adaptation
process takes place is frequency-dependent, we speculate that the corresponding time window, here
termed –mechanical echo–, is likely to increase with the intensity of the mechanical stimulus. We have
included this aspect in the discussion section (see lines 375-378).

o Comment 8

7-Line 47-50: Seems to require rephrasing.

8-Line 217: Figure reference is likely to 2F.

Response

We have corrected the typos accordingly.

4 References

1. Schaller, M. D. & Parsons, J. T. pp125 FAK -Dependent Tyrosine Phosphorylation of Paxillin
Creates a High-Affinity Binding Site for Crk . Mol. Cell. Biol. 15, 2635â2645 (1995).

2. Zhou, J. et al. BMP receptor-integrin interaction mediates responses of vascular endothelial

Smad1/5 and proliferation to disturbed flow. J. Thromb. Haemost. 11, 741â755 (2013).

3. Du, J. et al. Integrin activation and internalization on soft ECM as a mechanism of induction of
stem cell differentiation by ECM elasticity. Proc. Natl. Acad. Sci. 108, 9466â9471 (2011).

4. Kopf, J., Petersen, A., Duda, G. N. & Knaus, P. BMP2 and mechanical loading cooperatively
regulate immediate early signalling events in the BMP pathway. BMC Biol. 10, 37 (2012).

5. Brauer, E. et al. Collagen Fibrils Mechanically Contribute to Tissue Contraction in an In Vitro
Wound Healing Scenario. Adv. Sci. (Weinheim, Baden-Wurtemberg, Ger. 6, 1801780 (2019).

Version 1:

Reviewer comments:

Reviewer #3

(Remarks to the Author)

In their response Gorlitz et al have clarified all of my concerns. Therefore, I support the publication of the current manuscript. The authors have elucidated their choice of correlating focal adhesions with the changes in BMP signalling upon inhibition of the actin turnover using Jasplakinolide. Indeed, the strong entanglement between the tension in cytoskeletal networks and the adhesion complex remodelling makes independent measurements challenging. The authors have now addressed this connection clearly at several places in their manuscript and further strengthened the importance of an adaptation of the focal adhesions to the cyclic stimuli as opposed to a maturation per se which seems key to their data and interpretation. The agreement between Jas treatment, myosin inhibition (blebbistatin experiments shared in the response) and cells on soft substrates in terms of the reduced load-induced synergism with BMP signalling is very intriguing. In fact, the comment of similarity of the data presented here and to cells migrating on a non-linearly deformed soft collagen scaffold that undergo temporal changes by interacting with differently strained regions is very interesting. Presumably the oscillating focal adhesion turnover dynamics is what drives increased BMP signalling as opposed to uniform substrates (even if at different stiffness). The mechanical echo described in this manuscript might be long enough for the cells to cross a soft collagen patch and keep BMP signalling augmented.
